# Guijing2501 (*Citrus unshiu*) Has Stronger Cold Tolerance Due to Higher Photoprotective Capacity as Revealed by Comparative Transcriptomic and Physiological Analysis and Overexpression of Early Light-Induced Protein

**DOI:** 10.3390/ijms242115956

**Published:** 2023-11-03

**Authors:** Cui Xiao, Ligang He, Wenming Qiu, Zeqiong Wang, Xiujuan He, Yuxiong Xiao, Zhonghai Sun, Zhu Tong, Yingchun Jiang

**Affiliations:** Hubei Key Laboratory of Germplasm Innovation and Utilization of Fruit Trees, Institute of Fruit and Tea, Hubei Academy of Agricultural Sciences, Wuhan 430064, China; xiaocui@hbaas.com (C.X.); lghe44@aliyun.com (L.H.); qiuwm@hbaas.com (W.Q.); wangzq@hbaas.com (Z.W.); hexiujuan@hbaas.com (X.H.); yuxiongx@163.com (Y.X.); hbfruit@126.com (Z.S.)

**Keywords:** citrus, cold stress, early light-induced protein, transcriptome

## Abstract

Cold is one of the major limiting factors for citrus production, particularly extreme cold waves. Therefore, it is of great importance to develop cold-tolerant varieties and clarify their cold tolerance mechanisms in citrus breeding. In this study, comparative transcriptomic and physiological analyses were performed to dissect the cold tolerance mechanism of Guijing2501 (GJ2501), a new satsuma mandarin (*Citrus unshiu*) variety with about 1 °C lower LT50 (the median lethal temperature) relative to Guijing (GJ). The physiological analysis results revealed that GJ2501 is more cold-tolerant with less photoinhibition, PSII photodamage, and MDA accumulation, but higher POD activity than GJ under cold stress. Comparative transcriptomic analysis identified 4200 DEGs between GJ and GJ2501, as well as 4884 and 5580 up-regulated DEGs, and 5288 and 5862 down-regulated DEGs in response to cold stress in GJ and GJ2501, respectively. “Photosynthesis, light harvesting” and “photosystem” were the specific and most significantly enriched GO terms in GJ2501 in response to cold stress. Two *CuELIP1* genes (encoding early light-induced proteins) related to the elimination of PSII photodamage and photoinhibition were remarkably up-regulated (by about 1000-fold) by cold stress in GJ2501 as indicated by RT-qPCR verification. Overexpression of *CuELIP1* from GJ2501 in transgenic *Arabidopsis* protected PSII against photoinhibition under cold stress. Taken together, the cold tolerance of GJ2501 may be ascribed to its higher photoprotective capacity under cold stress.

## 1. Introduction

Citrus is one of the most widely cultivated fruit crops in the world. In 2019, the total production of citrus in China reached 43.54 million tons, accounting for about one third of the world’s total production (FAOSTAT); moreover, the production of mandarin (19 million tons) in China also ranked the first globally [1,2]. Citrus does not need low temperature to induce flowering and flourishing; instead, it is vulnerable to the negative impact of low temperature as a tropical or subtropical fruit tree and must endure seasonal extreme low temperatures during winter and early spring [1,3]. Besides, periodical extreme cold waves are becoming more frequent due to climate change, which have caused great losses in the citrus industry, particularly for the late-maturing citrus that needs to survive the winter in the northern fringe growing areas. The five devastating freezes of the 1980s in Florida resulted in an immense hit on 30% of Florida’s entire citrus industry and generational farms were put out of business [4]. In 1990, a cold wave caused about 500-million-dollar losses in citrus fresh fruit and affected about 450,000 ha of citrus trees in California [5], and in 2010, two consecutive frosts caused 142-million-euro losses for citrus industry of Valencia [1]. Moreover, the lengthy periods of snowing and freezing in southern China in 2008 also caused great economic losses for the citrus industry in China. Therefore, it is critical to breed citrus varieties of higher cold tolerance and take advantage of cold-tolerant rootstocks, clarify their cold tolerance mechanisms, and unravel the key genes for the cold tolerance.

*Poncirus trifoliata* is a classic extremely cold-hardy rootstock widely used for citrus production in China. The molecular mechanism underlying the cold tolerance of *P. trifoliata* has been extensively studied in the past decade. Low temperature responsive genes and changes in the transcriptome of *P. trifoliata* in response to cold stress have been initially characterized by suppression subtractive hybridization and RNA-sequencing [6,7], followed by the discovery of a series of cold-tolerant regulators and genes. The transcription factors *PtrbHLH* (Basic Helix-Loop-Helix) and *PtrERF109* (Ethylene-responsive factor 109) of *P. trifoliata* were identified to confer cold tolerance by modulating peroxidase (POD) and/or catalase (CAT) mediated H_2_O_2_ scavenging [8,9,10]. *PtrICE1* (inducer of CBF Expression 1) interacts with *PtADC* (arginine decarboxylase) and modulates the polyamine levels [11]; *PtrERF9* interacts with *PtrACS1* (ACC synthase1) and *PtrGSTU17* (glutathione S-transferase U17) to respectively regulate ethylene biosynthesis and ROS homeostasis [12]; and *PtrMYC2* regulates *PtrBADH-l* (betaine aldehyde dehydrogenase-like gene) and glycine betaine synthesis [13] to play a role in the cold tolerance of *P. trifoliata*. Moreover, *PtrBAM1* (β-Amylase 1) modulates the soluble sugar level [14]; *PtrERF108* regulates *PtrRafS* (raffinose synthase) and raffinose synthesis [15]; and *PtrAHL14/17-PtrHATs-PtrA/NINV7* module regulates sucrose catabolism [16], to contribute to the cold tolerance of *P. trifoliata*. Besides, the high-quality genome of *P. trifoliata* revealed that it is cold-tolerant due to specific adaptation in the C-repeat/DREB binding factor (CBF)-dependent and CBF-independent cold signaling pathways [17].

Recently, Chongyi wild mandarin, which is more cold-tolerant than *P. trifoliata*, was identified and its transcriptomic characteristics in response to cold were analyzed [18]. Carrizo citrange, a hybrid of *P. trifoliata* and sweet orange (*Citrus sinensis*) with certain cold tolerance derived from the *P. trifoliata* parent, is extensively used as rootstock in the Mediterranean region with extremely calcareous soil, whereas *P. trifoliata* is rarely used due to its sensitivity to iron chlorosis [1]. A recent study revealed the physiological characteristics of carrizo citrange under prolonged low temperature stress [19], and another study compared the transcriptomic characteristics under long-term low temperature stress between cold-tolerant carrizo citrange and cold-sensitive *C. macrophylla* as rootstock for grafting of Valencia delta seedless orange [20]. Besides, tetraploid carrizo citrange was identified as a rootstock to enhance natural cold tolerance of common clementine [21], and *P. trifoliata* rootstock obviously affects the expression of genes in satsuma mandarin scion during cold acclimation [22]. Actually, plant genotype is the determinant of the potential of cold tolerance, whereas other factors can only modify this potential [23]. According to previous studies of citrus, mandarin is the most cold-tolerant, followed by sweet orange and grapefruit, while lemon and lime are the least cold-tolerant [23,24], and substantial research has indicated that satsuma mandarin (*C. unshiu*) is the most cold-hardy commercial cultivar [23,24,25,26,27].

Guijing2501 satsuma mandarin (*C. unshiu*; GJ2501) (−9.5 °C LT50) is a bud mutation variety of Guijing satsuma mandarin (GJ) (−8.67 °C LT50) with about 1 °C lower LT50 (the median lethal temperature) [28]. It was selected from field survey during a hard freeze in 1977 in Hubei Province of China, and now is cultivated as a new cultivar in Shiyan city, a northern fringe region of citrus production [28]. In our previous study, the CuCBL6/8 (calcineurin B-like protein)–CuCIPK8/14 (CBL-interacting protein kinase) signaling networks were identified to be involved in the cold response of GJ2501 and overexpression of *CuCIPK16* from GJ2501 promoted the cold tolerance and POD activity while resulted in a lower malondialdehyde (MDA) content in transgenic *Arabidopsis* relative to the wild type [29]. However, the mechanism underlying the cold tolerance of GJ2501 remains to be further clarified. In this study, in order to uncover the potential cold tolerant genes and mechanisms in GJ2501, we compared the physiological traits and transcriptome characteristics between GJ and GJ2501, then, we selected 12 candidate up-regulated and down-regulated cold-responsive genes, respectively, for further verification by RT-qPCR analysis, finally, we cloned the early light-induced protein 1 from GJ2501 and overexpressed in *Arabidopsis* to analyze its function. The results in this study will provide valuable candidate genes for the breeding of cold-hardy citrus.

## 2. Results

### 2.1. Differences in Cold Stress Tolerance between GJ and GJ2501

To compare their cold tolerance, GJ and its cold-tolerant mutation variety GJ2501 were exposed to freezing stress (−4 °C) and chilling stress (4 °C). When exposed to freezing stress for 30 h and recovery at ambient temperature for 7 d, GJ and GJ2501 showed visible phenotypic differences. The leaves of GJ were severely damaged, whereas those of GJ2501 were only slightly injured, as indicated by noticeable decreases in chlorophyll fluorescence (Figure 1A) and Fv/Fm ratio (Figure 1B) in GJ relative to GJ2501. When exposed to chilling stress, although there were no visible phenotypic differences in chlorophyll fluorescence (Appendix A) and Fv/Fm ratio (Appendix A), GJ and GJ2501 showed obvious differences in MDA content (Figure 1C) and POD activity (Figure 1D). At the later stage (8 d and 16 d) of chilling stress, the MDA content in GJ2501 dramatically decreased to be substantially lower than that in GJ. The POD activity of GJ2501 consistently increased throughout the whole chilling stress process and was substantially higher than that of GJ at the later stage (4 d, 8 d and 16 d). Overall, the above results suggested that GJ2501 has stronger cold tolerance than GJ.

### 2.2. Transcriptome Profiles of GJ and GJ2501 during Time-Course Cold Stress

For elucidation of the potential molecular mechanism for the cold tolerance of GJ2501, GJ and GJ2501 were subjected to cold treatment for 0.5, 1, 2, 4, 8 and 16 d, followed by a comparative transcriptomic analysis. A total of 111.25 Gb clean data and an average of 2.14 Gb clean data per sample were generated, and the Q30 values (percentage of sequences whose sequencing error rate was lower than 0.1%) were 92.78–95.1% (Appendix A). These high-quality clean reads were then mapped to the reference genome (Citrus_sinensis v3.0), with the mapping rate ranging from 86.32% to 93.79% (Appendix A).

In the principal component analysis (PCA), the first principal component (PC1), which accounted for 33.41% of the total variance, could well separate the samples before cold stress (0 d) and at early cold stress stages (0.5 and 1 d) from the samples at later cold stress stages (2, 4, 8 and 16 d) (Figure 2A). PC2 (accounting for 17.98% of the total variance) could further separate the samples before cold stress (0 d) from those at early cold stress stages (0.5 and 1 d), and the samples at 2 d and 4 d from those at 8 d and 16 d of cold stress (Figure 2A). These results revealed that both GJ and GJ2501 were regulated by cold stress, particularly the samples at later stages of cold stress, as they were the most distinctly separated from those samples before cold stress.

Differentially expressed genes (DEGs) were screened between GJ and GJ2501 through comparison of their transcriptome data at each time point. Figure 2B shows that there were fewer DEGs (100 up-regulated and 43 down-regulated) before cold treatment (0 d), suggesting that GJ and GJ2501 have similar genetic background. However, after cold treatment, there were substantial increases in DEGs, especially at 1 d (852 up-regulated and 1247 down-regulated genes) and 16 d (605 up-regulated and 1269 down-regulated genes), suggesting that GJ and GJ2501 might have different cold stress responses.

The cold-responsive DEGs were further screened by comparing the transcriptome data between 0 d and each of other time points after cold treatment in GJ and GJ2501. Figure 2B reveals that both GJ and GJ2501 had substantial DEGs after cold treatment. Moreover, in both GJ and GJ2501, there were more up-regulated DEGs than down-regulated ones at the early stages of cold treatment; however, it was the opposite case at the later stages. These results indicated that both GJ and GJ2501 are regulated by cold stress. However, the number of DEGs first increased and then decreased in GJ under cold treatment, while consistently increased during the whole cold stress process in GJ2501. In addition, there were more DEGs in GJ2501 (6136) than in GJ (4798) at 16 d after cold treatment. GJ2501 had a larger total number of both up-regulated (5580) and down-regulated DEGs (5862) than GJ (4884 up-regulated and 5288 down-regulated DEGs) (Datasheet S1–S4). These results implied that the enhanced cold tolerance of GJ2501 might be attributed to a higher number of cold-responsive DEGs relative to GJ.

### 2.3. KEGG and GO Enrichment Analysis of DEGs between GJ and GJ2501

The DEGs between GJ and GJ2501 at each time point were merged into one gene set named as “GJ_GJ2501_DEG”, which comprised a total of 4200 DEGs (Datasheet S5). KEGG and GO enrichment analysis were then performed on the DEGs in this gene set. The results revealed that these DEGs were significantly enriched in a total of seven KEGG pathways, namely “Photosynthesis”, “Photosynthesis-antenna proteins”, “Carbon fixation in photosynthetic organisms” involved in energy metabolism, “Starch and sucrose metabolism”, “Amino sugar and nucleotide sugar metabolism”, “Galactose metabolism” involved in carbohydrate metabolism, and “Porphyrin and chlorophyll metabolism” related to the metabolism of cofactors and vitamins (Figure 3A and Appendix A). “Photosynthesis” was the most significantly enriched KEGG pathway (Figure 3A). Moreover, these DEGs were significantly enriched in 18 GO terms related to cellular components (CC), 9 GO terms associated with biological processes (BP), and 6 GO terms associated with molecular functions (MF) (Appendix A). About 55% of the significantly enriched GO terms belonged to CC, and all these terms had close association with photosynthesis including photosystem, chloroplast, and plastid (Appendix A). The numbers of DEGs in “plastid” and “chloroplast” were respectively 185 and 175, which were the top two enriched GO terms in CC (Appendix A). “Photosynthesis, light harvesting” was the most significantly enriched among all GO terms; “photosystem I” was the GO term with the most significant enrichment in CC; and “chlorophyll binding” was the most significantly enriched in MF (Figure 3B). These results implied that the difference in photosynthesis between GJ and GJ2501 might contribute to their differences in cold tolerance.

### 2.4. KEGG and GO Enrichment Analysis of Up-Regulated DEGs in GJ and GJ2501 during Cold Stress

The up-regulated cold-responsive DEGs at each time point in GJ and GJ2501 were merged into one gene set named as “GJ_up_DEG” and “GJ2501_up_DEG”, which included a total of 4884 DEGs and 5580 DEGs, respectively (Datasheet S1; Datasheet S2). KEGG and GO enrichment analysis were performed on the DEGs in these two gene sets, respectively. The results revealed that “Ribosome biogenesis in eukaryotes” was the significantly enriched KEGG pathway in both “GJ_up_DEG” and “GJ2501_up_DEG”, whereas “Photosynthesis-antenna proteins” and “Photosynthesis” were only significantly enriched in “GJ2501_up_DEG” (Figure 4A,C, Appendix A). According to the comparison of GO enrichment analysis, “photosynthesis, light harvesting”, “photosynthesis, light harvesting in photosystem I”, “photosystem II”, and “photosystem I” were significantly enriched only in “GJ2501_up_DEG” and were the top four significantly enriched GO terms (Figure 4B,D, Appendix A). These results suggested that the up-regulated cold-responsive DEGs in GJ2501 involved in the light harvesting of photosynthesis and photosystem I/II might have important contributions to its cold tolerance.

### 2.5. KEGG and GO Enrichment Analysis of Down-Regulated DEGs in GJ and GJ2501 during Cold Stress

The down-regulated cold-responsive DEGs at each time point in GJ and GJ2501 were merged into one gene set named as “GJ_down_DEG” and “GJ2501_down_DEG”, which included a total of 5288 DEGs and 5862 DEGs, respectively (Datasheet S3; Datasheet S4). KEGG and GO enrichment analysis were carried out on the DEGs in these two gene sets, respectively. The results revealed that “Photosynthesis-antenna proteins” and “Photosynthesis” were significantly enriched KEGG pathways in both “GJ_down_DEG” and “GJ2501_down_DEG” (Figure 5A,C, Appendix A). There were 19 and 30 significantly enriched GO terms in “GJ_down_DEG” and “GJ2501_down_DEG”, respectively. A comparison of the GO enrichment analysis results revealed that 73.7% of GO terms significantly enriched in “GJ_down_DEG” were shared by “GJ2501_down_DEG”. However, the GO terms “photosystem I”, “chloroplast/plastid stroma”, “photosynthesis, light harvesting”, “photosynthesis, light harvesting in photosystem I”, “monocarboxylic acid/carboxylic acid/organic acid/lipid biosynthetic process”, “dicarboxylic acid/fatty acid/cellular lipid/lipid metabolic process”, and “isomerase/glucosyltransferase/oxidoreductase activity” were significantly enriched only in “GJ2501_down_DEG” (Figure 5B,D, Appendix A). The GO terms “photosynthesis”, “small molecule catabolic process”, “auxin-activated signaling pathway”, “envelope”, and “organelle envelope” were significantly enriched only in “GJ_down_DEG” (Figure 5B,D, Appendix A). These results implied that the DEGs in these unique GO terms, respectively, in GJ and GJ2501 might also contribute to their differences in cold tolerance.

### 2.6. KEGG and GO Enrichment Analysis of WGCNA Modules with Different Module-Trait Correlation Patterns between GJ and GJ2501

A weighted gene co-expression network analysis (WGCNA) was conducted for the expression of all genes in GJ and GJ2501 at all the time points, and 13 WGCNA modules showing 13 gene expression patterns under time-course cold stress were identified (Figure 6). The turquoise, blue, brown, and yellow modules ranked within the top four in gene number (Figure 6B). According to the module-trait relationships (Appendix A), the modules (pink, blue, green, tan, magenta, and grey) with different module-trait correlation patterns between GJ and GJ2501 were clustered together (Figure 7A). The genes in these modules were merged into one gene set named as “different_module”, which comprised a total of 4760 genes (Datasheet S6).

KEGG and GO enrichment analysis were performed on the “different_module” gene set. As a result, the significantly enriched KEGG pathways and GO terms in “different_module” were quite similar to those in “GJ_GJ2501_DEG”. The KEGG pathways “Photosynthesis”, “Photosynthesis-antenna proteins”, and “Carbon fixation in photosynthetic organisms” involved in energy metabolism, and “Glyoxylate and dicarboxylate metabolism” and “Fructose and mannose metabolism” involved in carbohydrate metabolism were significantly enriched, particularly “Photosynthesis” (Figure 7B and Appendix A). Most of the significantly enriched GO terms were CC, and all these terms except for “plasma membrane” were closely related to photosynthesis including photosystem, chloroplast, and plastid (Appendix A). “Chlorophyll binding” was the only GO term related to MF and ranked the second in all significantly enriched GO terms (Figure 7C). “Photosynthesis, light harvesting” was the most significantly enriched among the GO terms of BP (Figure 7C). These results further implied that the difference in photosynthesis between GJ and GJ2501 might contribute to their different cold tolerance.

### 2.7. KEGG and GO Enrichment Analysis of WGCNA Modules with Similar Module-Trait Correlation Patterns between GJ and GJ2501

According to the module-trait relationships (Appendix A), the modules (black, greenyellow, red, yellow, purple, brown, and turquoise) with similar module-trait correlation patterns between GJ and GJ2501 were clustered together (Figure 8A). The genes in these modules were merged into one gene set designated as “similar_module”, which comprised a total of 7750 genes (Datasheet S7). KEGG and GO enrichment analysis on “similar_module” revealed that only one KEGG pathway, “Ribosome biogenesis in eukaryotes” shared by “GJ_up_DEG” and “GJ2501_up_DEG”, was significantly enriched in this gene set (Appendix A, Figure 8B). Besides, a total of 95 GO terms, including 22 associated with CC, 55 associated with BP, and 18 associated with MF were significantly enriched in this gene set (Appendix A, Figure 8C). These results indicated that there are some common KEGG pathways and GO terms involved in cold stress response in GJ and GJ2501.

### 2.8. Verification of Candidate Up-Regulated Cold-Responsive DEGs in GJ and GJ2501 by RT-qPCR Analysis

GJ2501 showed higher photosynthetic capacity than GJ as indicated by its stronger chlorophyll fluorescence and higher Fv/Fm ratio under cold stress. KEGG enrichment analysis on “GJ_GJ2501_DEG”, “GJ2501_up_DEG”, and “different_module” all suggested that “Photosynthesis” is the most significantly enriched KEGG pathway. The corresponding GO enrichment analysis further revealed that “photosynthesis, light harvesting” and “chlorophyll binding” were GO terms with the most significant enrichment respectively in BP and MF. Most of the significantly enriched GO terms of CC were chloroplast and chloroplast-related cellular components. Therefore, seven up-regulated cold-responsive DEGs possibly localized in the chloroplast (predicted by Plant-mPLoc software (Version 2.0) and Swiss-Prot/NR annotation, Appendix A) were selected for RT-qPCR verification. Other five genes respectively encoding galactinol synthase 2, ubiquinol oxidase 1a, CBL-interacting serine/threonine-protein kinase 9, cold-regulated 413 plasma membrane protein 1, and stress-induced KIN2-like protein were verified at the same time. The results demonstrated that all these 12 genes were substantially up-regulated by cold stress in both GJ and GJ2501; however, their expression levels in GJ2501 were always significantly higher than those in GJ (Figure 9). Notably, two genes encoding early light-induced protein 1, which contained the chlorophyll A-B binding domain and might participate in light harvesting of photosynthesis, were remarkably up-regulated under cold stress, and their expression was increased to the highest by up to about 1000 folds in GJ2501 (Figure 9). The enhanced expression of the 12 up-regulated cold-responsive DEGs in GJ2501 relative to GJ might contribute to the cold tolerance of GJ2501. However, their specific functions need to be further studied.

### 2.9. Verification of Candidate Down-Regulated Cold-Responsive DEGs in GJ and GJ2501 by RT-qPCR Analysis

The above KEGG enrichment analysis has indicated that “Photosynthesis” and “Photosynthesis-antenna proteins” were common significantly enriched KEGG pathways of down-regulated cold-responsive DEGs in GJ and GJ2501. At the later cold treatment stages, there were substantial down-regulated DEGs in both GJ and GJ2501, and many KCS (3-ketoacyl-CoA synthase) family members involved in fatty acid elongation were remarkably down-regulated. Therefore, seven down-regulated DEGs related to photosynthesis with predicted chloroplast subcellular localization and five KCS family members were selected for further RT-qPCR verification. The results revealed that all these 12 genes were substantially down-regulated at 16 d after cold stress in both GJ and GJ2501, and the expression levels of six genes (*CuPRK*, *CuRbcS*, *CuKCS1*, *CuKCS4*, *CuKCS19*, *CuKCS6*) in GJ2501 were significantly higher than those in GJ (Figure 10). It is noteworthy that *CuKCS19* and *CuKCS6* were, respectively, down-regulated by 100 and 345 folds in GJ, whereas only by 33 and 62 folds in GJ2501 at 16 d after cold stress (Figure 10). It remains to be determined whether the different expression levels of these genes between GJ and GJ2501 contribute to their different cold tolerance.

### 2.10. Overexpression of CuELIP1a Enhances the Cold Tolerance of Transgenic Arabidopsis

The early light-induced protein 1 (*CuELIP1a*), whose expression level was increased by up to 1000 folds in GJ2501, was further cloned and then overexpressed in *Arabidopsis* to analyze its function. A cold tolerance assay was carried out on the T3 generation transgenic *Arabidopsis* lines. Thirty-day-old *Arabidopsis* plants were subjected to 4 °C treatment for one month. After treatment, the wild type (WT) leaves exhibited severe damage and turned deep purple or yellow, whereas the leaves of the transgenic lines remained green (Figure 11A). Chlorophyll fluorescence determination revealed that the WT had noticeable decreases in chlorophyll fluorescence (Figure 11B) and Fv/Fm ratio (Figure 11C) relative to the transgenic lines. A further comparison between the WT and transgenic lines showed that the transgenic plants had significantly higher POD activity but lower MDA content than the WT (Figure 11D,E). Taken together, these results indicated that overexpression of *CuELIP1a* from GJ2501 improved cold tolerance of *Arabidopsis* leading to enhanced POD activity and decreased MDA content.

## 3. Discussion

It is well known that photosynthesis is a temperature-sensitive physiological process in plants, and cold stress will make photosystem II prone to photoinhibition with lower Fv/Fm ratios [1,30], as well as cause oxidative damages and activate antioxidant activities [1]. There has been substantial research on citrus about the correlation between cold tolerance and physiological traits. For example, overexpression of *PtrA/NINV* (localized in chloroplast and mitochondria) from *P. trifoliata* in tobacco enhanced its cold tolerance by promoting the photosynthesis efficiency [31]. A decrease in photosynthesis, lower Fv/Fm ratio, and degradation of chlorophylls a and b were observed in carrizo citrange under cold stress [19], whereas clementine grafted on tetraploid carrizo citrange rootstocks exhibited a less dramatic decrease in net photosynthesis (Pn) and chlorophyll fluorescence (Fv/Fm), as well as a lower level of MDA than that grafted on diploid carrizo citrange rootstocks under natural chilling stress [21]. Besides, triploid citrus varieties with a higher photosynthetic capacity are more cold-tolerant than their parents (diploid citrus) [32]. Moreover, the allopolyploid hybrid has lower photoinhibition (Fv/Fm), less oxidative marker accumulation and greater increases in antioxidant activities under cold and light stress relative to its diploid and doubled-diploid parents [33]. Mandarin is the most cold-tolerant citrus type, showing the least significant down-regulation of photosynthetic parameters and highly efficient antioxidant system under natural chilling stress [34], and *C. unshiu* shows the highest cold tolerance (−8.4 LT50) with a higher chlorophyll content and lower lipid peroxidation compared with *C. sinensis*, *C. paradise*, and *C. limon* [26,27].

In the present study, cold-tolerant GJ2501 had less dramatic decreases in chlorophyll fluorescence and Fv/Fm ratio, lower accumulation of MDA, but higher POD activity relative to GJ under cold stress. Further transcriptomic analysis revealed that “Photosynthesis” was the most significantly enriched KEGG pathway, and “photosynthesis, light harvesting”, “chlorophyll binding”, and “chloroplast and chloroplast-related cellular components” were the most significantly enriched GO terms, which is supported by the KEGG and GO enrichment analysis results of “GJ_GJ2501_DEG”, “GJ2501_up_DEG”, and “different_module”. Seven substantially up-regulated cold-responsive chloroplastic DEGs with higher expression in GJ2501 relative to GJ were further identified by RT-qPCR verification. All the results suggested that photosynthesis possibly plays a critical role in the cold tolerance of GJ2501.

Early light-induced proteins (ELIPs) are light-harvesting chlorophyll a/b-binding proteins, which are located in thylakoid membranes and act as photoprotectants to protect the chloroplast from photodamage during high-light stress [35,36,37,38]. In *Arabidopsis*, *AtELIP2* is activated by various environmental stresses associated with photoinhibition [36], and the accumulation of *AtELIP1* has a high correlation with the degree of PSII photodamage [37]. Overexpression of *ScELIPs* from *Syntrichia caninervis* could protect PSII from photoinhibition and stabilize the leaf chlorophyll content in transgenic *Arabidopsis* under high-light stress [38]. Besides, *RcELIPs* from *Rhododendron catawbiense* were substantially up-regulated during cold acclimation [39]. Overexpression of *MfELIP* from *Medicago sativa* ssp. *falcata* increased the tolerance of tobacco to abiotic stresses including chilling, freezing, high light, and osmotic stress [40]. In the present study, the photoinhibition and PSII photodamage of GJ2501 were less severe than those of GJ under cold stress, as indicated by the less dramatic decreases in chlorophyll fluorescence and Fv/Fm ratio. Then, the further transcriptomic analysis revealed that light harvesting of photosynthesis, photosystem I/II, and chlorophyll binding were the specific significantly enriched GO terms for the up-regulated cold-responsive DEGs in GJ2501 and DEGs between GJ and GJ2501. Two *CuELIP1* genes related to the elimination of PSII photodamage and photoinhibition were remarkably up-regulated (up to about 1000 folds) by cold stress in GJ2501. Overexpression of *CuELIP1* from GJ2501 protected PSII against photoinhibition in transgenic *Arabidopsis* under cold stress. All these results implied that the enhanced photoprotective capacity conferred by *CuELIP1* in GJ2501 may contribute to its cold tolerance. However, the mechanism needs to be further clarified.

Ten other DEGs, including *CuATFPa/b* (acyltransferase), *CuERD7* (the early response to dehydration 7), *CuGOLS2* (galactinol synthase 2), *CuAOX1A* (alternative oxidase 1a), *CuCOR413PM1* (cold-regulated 413 plasma membrane protein 1), *CuCIPK16* (CBL-interacting protein kinase 16), *CuFRI3* (ferritin-3), *CuKIN2-like* (stress-induced KIN2-like protein), and *CuUKP* (unknown protein), were substantially up-regulated by cold stress, and their expression in GJ2501 was significantly higher than that in GJ. In *Arabidopsis*, *AtATFP* participates in the synthesis of fatty acid phytyl ester to maintain the integrity of thylakoid membranes in chloroplasts under abiotic stress [41], and *AtERD7* is associated with remodeling of membrane lipid composition during cold stress [42]. Overexpression of *ScGolS1* from *Solanum commersonii* conferred freezing tolerance to transgenic potato [43], and overexpression of *CsGolS1* from *Cucumis sativus* enhanced the assimilate translocation efficiency under cold stress in cucumber [44]. AOX^K^ is a natural SNP variant of AOX conferring cold tolerance to watermelon [45], and the expression of *AOX2a* was regulated by cold stress in carrot [46]. The expression of *PtrCOR413 IM1* was higher in cold-hardy *P. trifoliata* than that of *CmCOR413 IM1* in cold-sensitive *C. macrophylla* under low temperatures [47]. In our previous study, overexpression of *CuCIPK16* (homologue of *AtCIPK9*) from GJ2501 enhanced the cold tolerance of transgenic *Arabidopsis* [29]. To date, there has been rare research on ferritin-3 and stress-induced KIN2-like protein associated with cold stress. The functions of the above genes including the unknown protein correlated with the cold tolerance of GJ2501 will be clarified in our future research.

## 4. Materials and Methods

### 4.1. Plant Materials and Cold Tolerance Assay

GJ satsuma mandarin (*C. unshiu*) is a commercial citrus cultivar and GJ2501 is a new citrus cultivar which was selected by our group. Both the two plant materials were kept at the Institute of Fruit and Tea, Hubei Academy of Agricultural Sciences, Wuhan, China. One-year-old uniform and healthy bud-grafting satsuma mandarin plants (GJ and GJ2501) were cultivated under greenhouse conditions with a 16 h light/8 h dark photoperiod at 25 °C and routinely pruned for the stimulation of new leaf growth. For the cold treatment assay, the plants were kept in a low-temperature (−10~50 °C) growth chamber (Ningbo le Electrical Instrument Manufacturer Co., Ltd. [Ningbo, China], RLD-1000E-4DW) with a 16 h light (10,000 lux)/8 h dark (0 lux) photoperiod. As for freezing stress, −4 °C exposure was performed on the plants for 0, 6 and 30 h, followed by 7 d of recovery at 25 °C. As for chilling stress, 4 °C exposure was performed on the plants for 0, 0.5, 1, 2, 4, 8 and 16 d. An IMAGING-PAM chlorophyll fluorimeter (MAXI version, Heinz Walz GmbH, Effeltrich, Germany) was employed to measure the chlorophyll fluorescence and Imaging Win v2.46i software supplied with the system was used to calculate the Fv/Fm ratio. After cold treatment, the leaves were collected to measure the MDA content and POD activity with relevant detection kits (A003-1-2 for MDA, A084-3-1 for POD, Nanjing Jiancheng Bioengineering Institute, Nanjing, China).

### 4.2. RNA-Sequencing

The leaves after 4 °C treatment at 0, 0.5, 1, 2, 4, 8 and 16 d were sampled for further RNA-sequencing. Total RNA was extracted using an RNAprep Pure Plant Plus Kit (Polysaccharides&Polyphenolics-rich) (DP441, TIANGEN, Beijing, China) and the RNA integrity and concentration were checked using an Agilent 2100 Bioanalyzer (Santa Clara, CA, USA). The mRNA was isolated by NEBNext Poly (A) mRNA Magnetic Isolation Module (E7490, NEB, Ipswich, MA, USA). The cDNA library was constructed by NEBNext mRNA Library Prep Master Mix Set for Illumina (E6110, NEB, Ipswich, MA, USA). The constructed cDNA libraries were sequenced by the Illumina HiSeq™ 2500 platform (San Diego, CA, USA). Low quality reads, such as only adaptor, unknown nucleotides > 5%, or Q20 < 20% (percentage of sequences with sequencing error rates < 1%), were removed by perl script. Clean reads that were filtered from the raw reads were mapped to sweet orange genome (Citrus_sinensis v3.0) using Hisat2 software (Version 2.1.0) [48]. The levels of gene expression were determined using TPM values with the RSEM software (Version 1.3.3) [49]. Principal component analysis (PCA) was employed through the Majorbio Cloud Platform https://www.majorbio.com (accessed on 26 October 2016) to clarify the correlation and variation between samples. Differential expression was analyzed using DESeq 2 (Version 1.24.0) with “Fold change ≥ 2” and “*p*-adjust < 0.05” [50].

### 4.3. KEGG and GO Enrichment Analysis of Specific Gene Sets

The DEGs between GJ and GJ2501 at each time point were merged into one gene set and named as “GJ_GJ2501_DEG”. The up-regulated and down-regulated cold-responsive DEGs at each time point in GJ were merged into one gene set, which was named as “GJ_up_DEG” and “GJ_down_DEG”, respectively. Similarly, the up-regulated and down-regulated cold-responsive DEGs at each time point in GJ2501 were merged into one gene set, which was designated as “GJ2501_up_DEG” and “GJ2501_down_DEG”, respectively. WGCNA was performed through the Majorbio Cloud Platform https://www.majorbio.com (accessed on 26 October 2016). The genes in WGCNA modules with different and similar module-trait correlation patterns between GJ and GJ2501 were merged into one gene set, which was respectively named as “different_module” and “similar_module”. KEGG and GO enrichment analysis were respectively performed on the above seven gene sets by KOBAS (Version 2.1.1) [51] and GOATOOLS (Version 0.6.5) [52], and the *p*-value was Bonferroni-corrected. The top 20 significantly enriched (*p*-adjust < 0.05) KEGG pathways and GO terms are presented in the bubble chart.

### 4.4. RT-qPCR Verification of Candidate DEGs

Samples at 0, 1, 4, and 16 d after cold treatment in GJ and GJ2501 were subjected to RT-qPCR analysis with the specific primers designed by the Primer Premier 5.0 software (Appendix A) [53]. RNA extraction and RT-qPCR were carried out following previous descriptions [29]. Each RT-qPCR pattern was verified with three biological replicates, and the experiment was repeated once. *CuEF1α* and samples at 0 d in GJ were taken as the internal reference for normalizing the expression level. The relative gene expression was calculated by the 2^−∆∆CT^ method [54]. Statistical analysis was conducted using Student’s *t* test and the asterisk indicates statistical significance (* *p* < 0.05; ** *p* < 0.01; *** *p* < 0.001).

### 4.5. Cold Tolerance Verification of Arabidopsis Overexpressing CuELIP1a

*CuELIP1a* was cloned and inserted into the pK7YWG2 vector by Gateway Clonase Enzyme (Invitrogen, Carlsbad, CA, USA) followed by transformation into *Agrobacterium tumefaciens* strain GV3101, according to the manufacturer’s instructions. The floral dip method was used to carry out *Arabidopsis* transformation as previously described [55]. Positive transformants were selected with the MS medium containing 50 μg mL^−1^ kanamycin, and then genomic PCR and qRT-PCR confirmation was carried out. Thirty-day-old *Arabidopsis* plants (T3 generation) were subjected to 4 °C treatment for one month, and then the phenotype, chlorophyll fluorescence, Fv/Fm ratio, MDA content and POD activity were compared between the overexpression lines (OE) and wild type (WT).

## 5. Conclusions

Our work revealed that GJ2501 is more cold-tolerant than GJ, as evidenced by its lower photoinhibition and PSII photodamage, lower accumulation of MDA content, but higher POD activity under cold stress. The stronger cold tolerance of GJ2501 may be attributed to its higher photoprotective capacity under cold stress. This speculation is supported by that facts that the specific GO terms related to light harvesting of photosynthesis and photosystem were the most significantly enriched in GJ2501, the two *CuELIP1* genes related to the elimination of PSII photodamage and photoinhibition were significantly up-regulated in GJ2501, and transgenic *Arabidopsis* overexpressing *CuELIP1a* from GJ2501 showed less photoinhibition and PSII photodamage and enhanced cold tolerance under cold stress leading to enhanced POD activity and decreased MDA content relative to WT. The candidate genes for the cold tolerance screened in this study will provide valuable gene resources for cold-hardy citrus breeding.

## Figures and Tables

**Figure 1 ijms-24-15956-f001:**
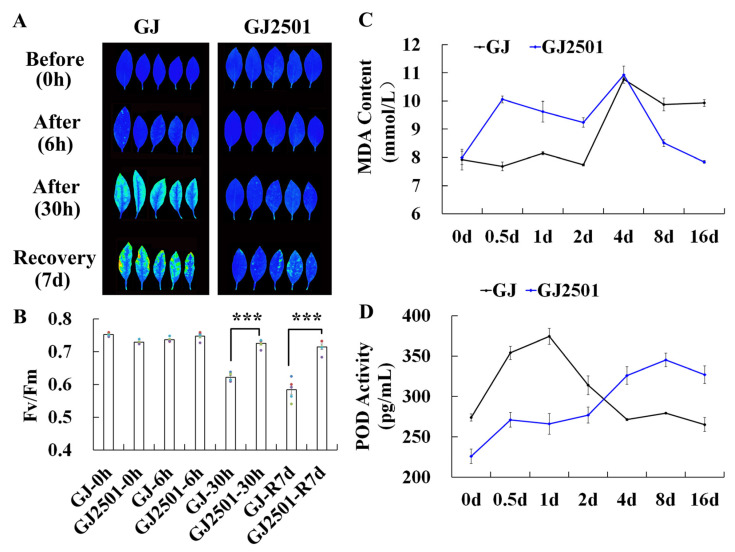
Images summarizing the difference in cold tolerance between GJ and GJ2501. The chlorophyll fluorescence image was photographed before −4 °C treatment (0 h), after −4 °C treatment for 6 h and 30 h, and recovery at 25 °C for 7 d (**A**), and the Fv/Fm ratio corresponding to each time point was calculated (**B**). The MDA content (**C**) and POD activity (**D**) were calculated after chilling stress. *** in (**B**) stands for statistical significance (*p* < 0.001, Student’s *t* test) and the colored dots in (**B**) stand for the individual data point.

**Figure 2 ijms-24-15956-f002:**
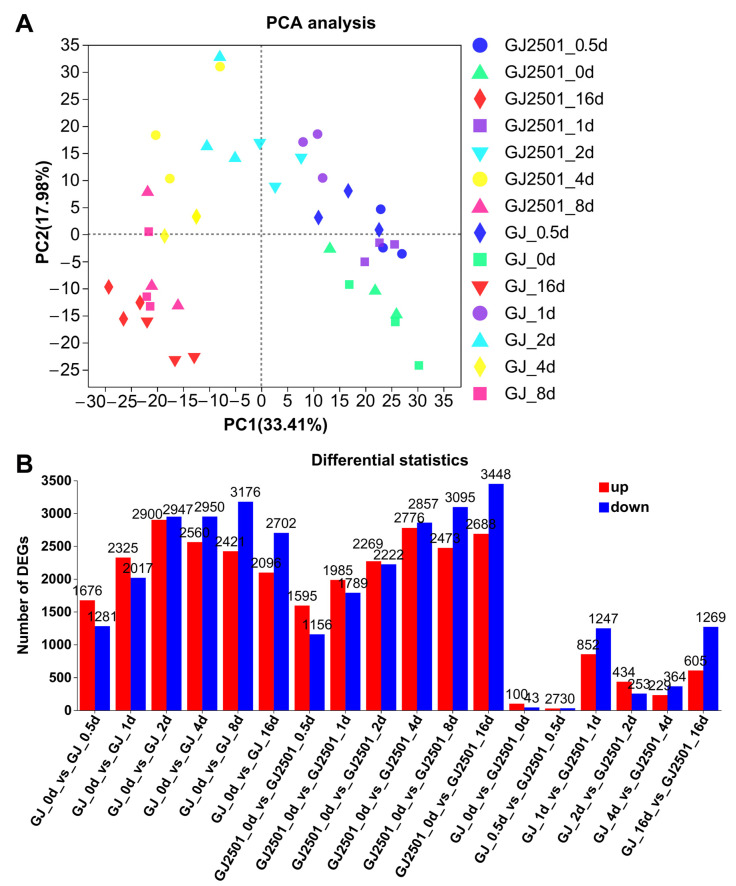
Principal component analysis (PCA) (**A**) and number of DEGs during time-course cold stress (**B**).

**Figure 3 ijms-24-15956-f003:**
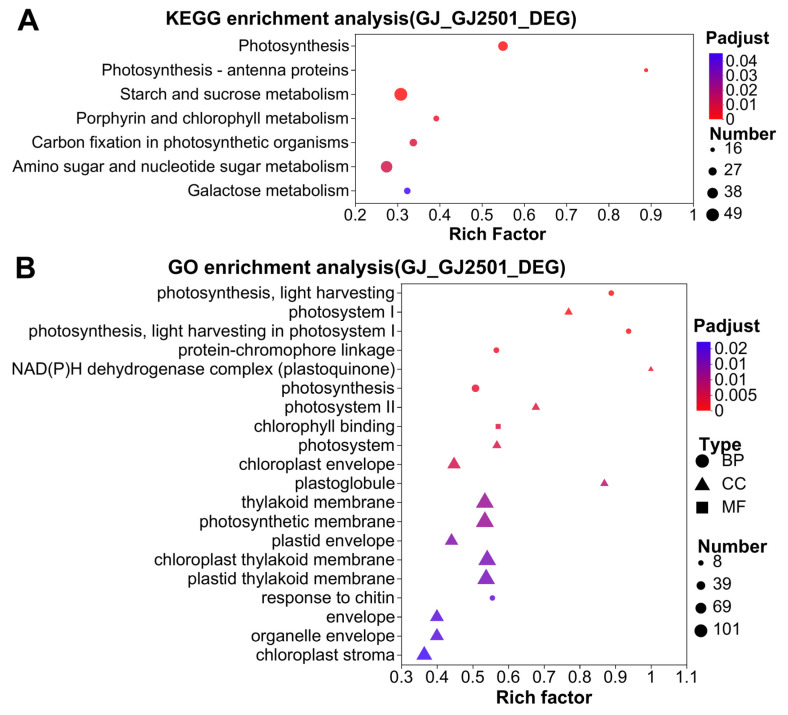
KEGG (**A**) and GO (**B**) enrichment analysis for the “GJ_GJ2501_DEG” gene set.

**Figure 4 ijms-24-15956-f004:**
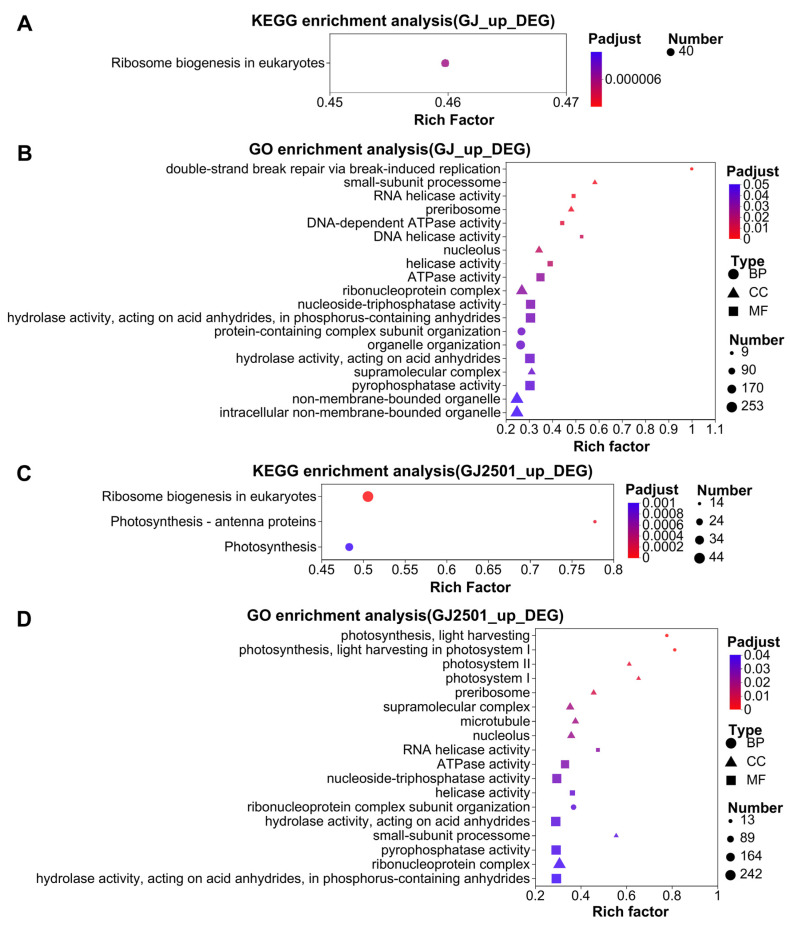
KEGG enrichment analysis on the “GJ_up_DEG” (**A**) and “GJ2501_up_DEG” (**C**) gene set; GO enrichment analysis on “GJ_up_DEG” (**B**) and “GJ2501_up_DEG” (**D**).

**Figure 5 ijms-24-15956-f005:**
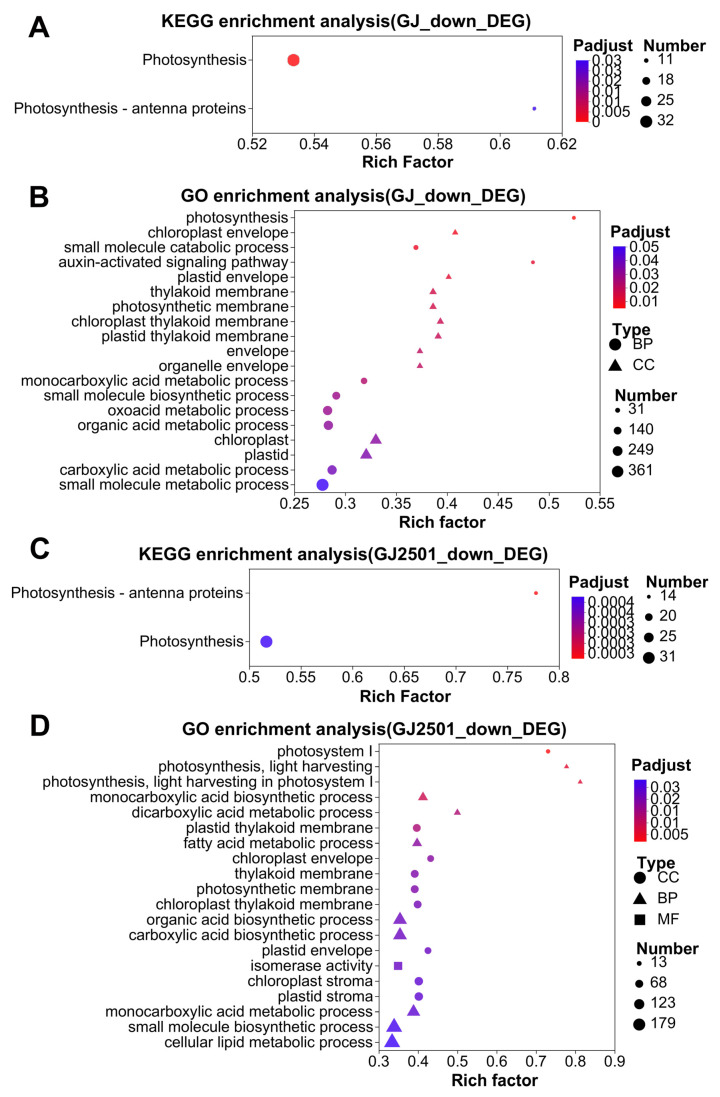
KEGG enrichment analysis on the “GJ_down_DEG” (**A**) and “GJ2501_down_DEG” (**C**) gene set; GO enrichment analysis on “GJ_down_DEG” (**B**) and “GJ2501_down_DEG” (**D**).

**Figure 6 ijms-24-15956-f006:**
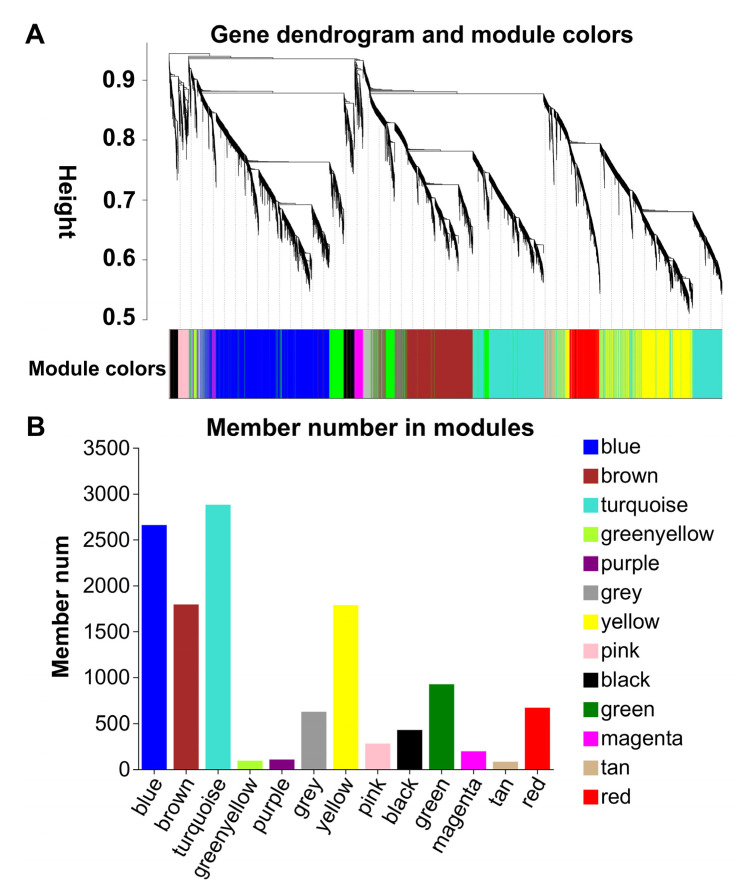
Dendrogram (**A**) and gene number (**B**) of the WGCNA modules.

**Figure 7 ijms-24-15956-f007:**
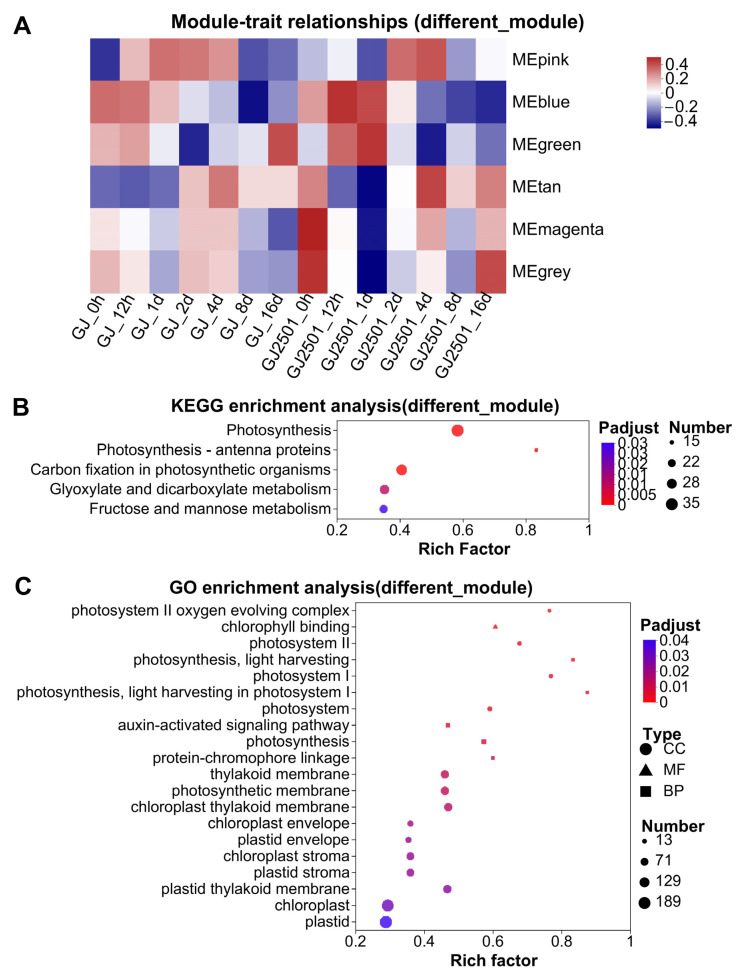
Module-trait relationships (**A**), KEGG enrichment analysis (**B**), and GO enrichment analysis (**C**) of WGCNA modules with different module-trait correlation patterns between GJ and GJ2501.

**Figure 8 ijms-24-15956-f008:**
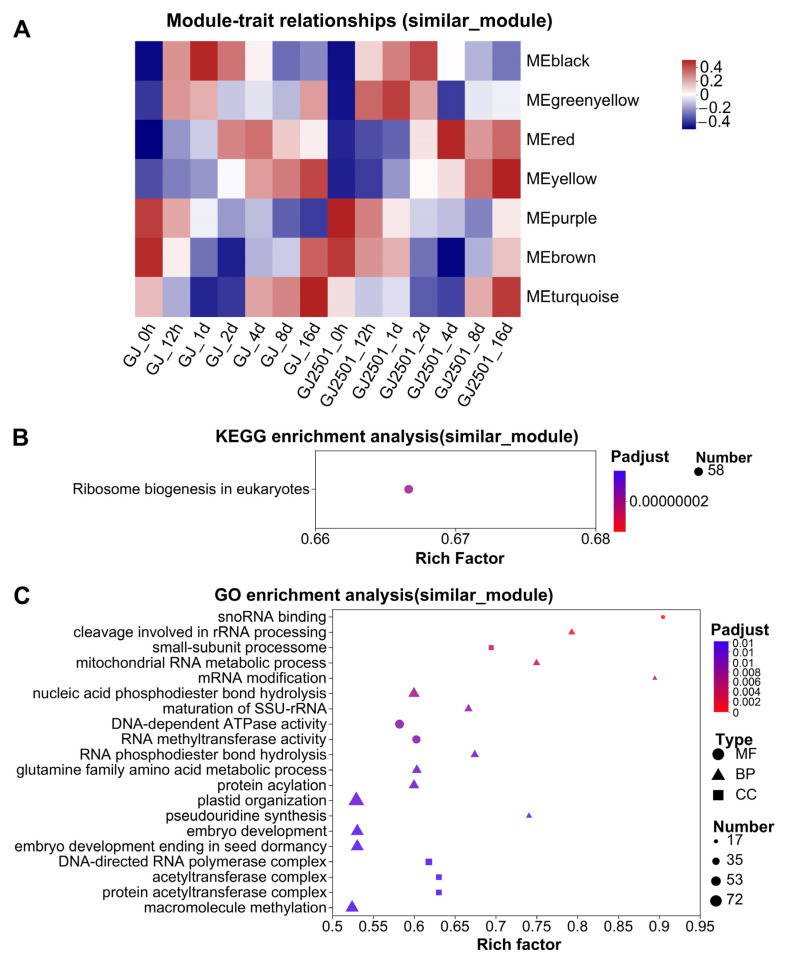
Module-trait relationships (**A**), KEGG enrichment analysis (**B**), and GO enrichment analysis (**C**) of WGCNA modules with similar module-trait correlation patterns between GJ and GJ2501.

**Figure 9 ijms-24-15956-f009:**
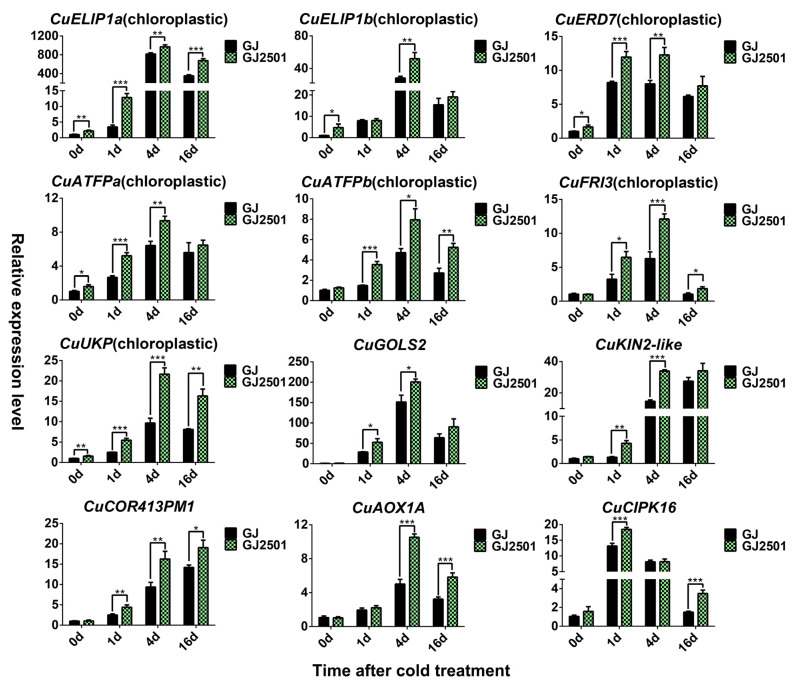
RT-qPCR analysis of the candidate up-regulated cold-responsive DEGs in GJ and GJ2501. Error bars represent inferential statistics according to the SE. * stands for statistical significance (*, **, and *** represent *p* < 0.05 *p* < 0.01, and *p* < 0.001, respectively, Student’s *t* test).

**Figure 10 ijms-24-15956-f010:**
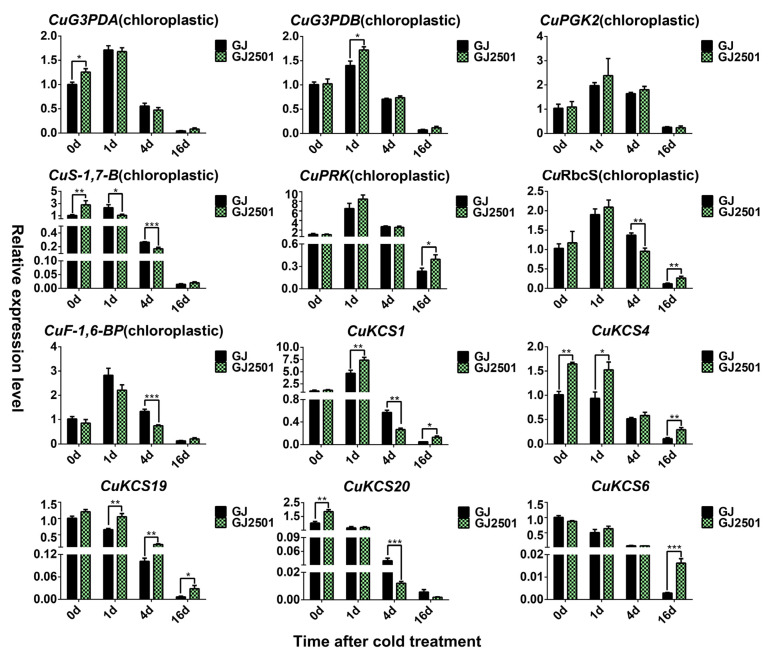
RT-qPCR analysis of the candidate down-regulated cold-responsive DEGs in GJ and GJ2501. Error bars represent inferential statistics according to the SE. * stands for statistical significance (*, **, and *** represent *p* < 0.05 *p* < 0.01, and *p* < 0.001, respectively, Student’s *t* test).

**Figure 11 ijms-24-15956-f011:**
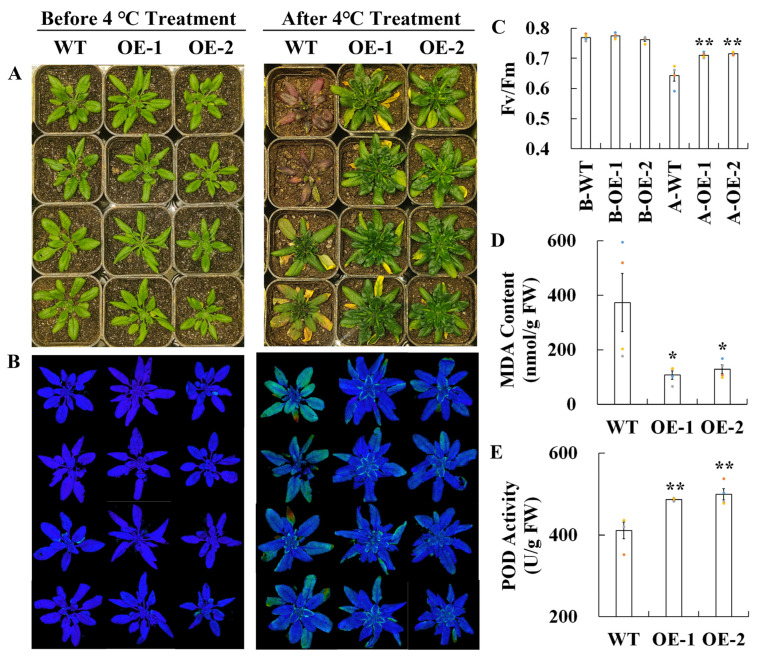
Comparison of phenotype (**A**), chlorophyll fluorescence (**B**), Fv/Fm ratio (**C**), MDA content (**D**), and POD activity (**E**) between WT and transgenic *Arabidopsis* lines under cold treatment (one month at 4 °C). The colored dots in (**C**–**E**) stand for the individual data point. Error bars in (**C**–**E**) represent inferential statistics according to the SE. * stands for statistical significance (* and ** represent *p* < 0.05 and *p* < 0.01, respectively, Student’s *t* test) relative to WT.

## Data Availability

Not applicable.

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
