# Peer review of "Guijing2501 (Citrus unshiu) Has Stronger Cold Tolerance Due to Higher Photoprotective Capacity as Revealed by Comparative Transcriptomic and Physiological Analysis and Overexpression of Early Light-Induced Protein"

_ijms, 2023, doi:10.3390/ijms242115956_

Round 1
Reviewer 1 Report
Comments and Suggestions for Authors
Dear Authors,
Reviewer comments ijms-2621443
The manuscript entitled „Guijing2501 (Citrus unshiu) has stronger cold tolerance due to higher photoprotective capacity as revealed by comparative transcriptomic and physiological analysis and CuELIP1a overexpression“ represents a useful study aimed at an investigation of enhanced cold tolerance in Satsuma mandarin variety Guijing GJ and its bud mutant GJ2501 with enhanced cold tolerance (about 1 °C lower LT50 than its parent) based on transcriptome analysis by RNA-seq approach validated by qRT-PCR of selected cold-responsive transcripts and selected physiological characteristics including chlorophyll fluorescence (photodamage), malondialdehyde (MDA) content and peroxidase (POD) activity. Moreover, overexpression of CuELIP1 from GJ2501 in transgenic Arabidopsis improved cold tolerance of the transformants due to enhanced photoprotection againts photoinhibition.
I can recommend the present manuscript for publication in International Journal of Molecular Sciences. However, I have some major comments on the present manuscript which are given below:
Materials and methods:
The source of both Satsuma mandarin genotypes GJ and GJ2501 used for the study has to be specified in Materials and methods.
Regarding both the chilling (+4 °C for 0-16 days) and freezing (-4 °C for 0-30 hours) treatments, the kind of growth chambre used and other treatment conditions such as light irradiance and photoperiod applied for the chilling and freezing treatments have to be added.
In Results, LT50 values are given for GJ and GJ2501 genotypes; however, no description of LT50 values determination is provided in Materials and methods. I thus think that a brief description of LT50 determination has to be added in Materials and methods.
For all software kinds mentioned in Materials and methods, e.g., Kobas or Goatools software, software version and web address or any other relevant reference have to be given.
Conclusion: I think that the main message of the present study, i.e., how CuELIP1a overexpression contributes to enhanced cold tolerance of transgenic Arabidopsis should be added AS Figure 12 to the manuscript.
Formal comments on the text:
Introduction, line 43: Remove the words „and the citrus industry“ in the statement: „…which have caused great losses in the citrus industry, particularly for the late-maturing citrus that needs to survive the winter in the northern fringe growing areas.“
Results, line 160: Add the word „enhanced“ prior to „cold tolerance“ in the statement „These results implied that the enhanced cold tolerance of GJ2501 might be attributed to a higher number of cold-responsive DEGs relative to GJ.“
Results, line 173: Correct the typing error in the term „chlorophyll“ (not „chlorophyII“).
Results, line 228: Add a comma both before and after the word „respectively“ in the statement „These results implied that the DEGs in these unique GO terms, respectively, in GJ and GJ2501might also contribute to their differences in cold tolerance.“
Line 298: Replace the word „higher“ with ůenhanced“ in the statement „The enhanced expression of 12 up-regulated cold-responsive DEGs…“
Line 317: Add a comma both preceding and following the word „respectively“ in the statment „I tis noteworthy that CuKCS19 and CuKCS6 were, respectively, down-regulated by 100 and 845 folds in GJ…“
Line 336: Modify the statement as follows: „Taken together, these results indicated that overexpression of CuELIP1a from GJ2501 improved cold tolerance of Arabidopsis leading to enhanced POD activity and decraesed MDA content.“
Line 366: Replace „less“ with „lower“ in the statement „lower accumulation of MDa“.
Line 395: Replace the word „higher“ with „enhanced“ in the statement „…that enhzanced photoprotective capacity conferred by cuELIP1 in GJ2501 may contribute to its cold tolerance.“
Conclusion, line 474: Replace the word „less“ with „lower“ in the statement „Our work revealed that GJ2501 is more cold-tolerant than GJ, as evidenced by its lower photoinhibition and PSII photodamage, lower accumulation of MDA, but higher POD activity…“
Final recommendation: Reconsider after a major revision.
Comments on the Quality of English Language
Dear Authors,
Reviewer comments ijms-2621443
The manuscript entitled „Guijing2501 (Citrus unshiu) has stronger cold tolerance due to higher photoprotective capacity as revealed by comparative transcriptomic and physiological analysis and CuELIP1a overexpression“ represents a useful study aimed at an investigation of enhanced cold tolerance in Satsuma mandarin variety Guijing GJ and its bud mutant GJ2501 with enhanced cold tolerance (about 1 °C lower LT50 than its parent) based on transcriptome analysis by RNA-seq approach validated by qRT-PCR of selected cold-responsive transcripts and selected physiological characteristics including chlorophyll fluorescence (photodamage), malondialdehyde (MDA) content and peroxidase (POD) activity. Moreover, overexpression of CuELIP1 from GJ2501 in transgenic Arabidopsis improved cold tolerance of the transformants due to enhanced photoprotection againts photoinhibition.
I can recommend the present manuscript for publication in International Journal of Molecular Sciences. However, I have some major comments on the present manuscript which are given below:
Materials and methods:
The source of both Satsuma mandarin genotypes GJ and GJ2501 used for the study has to be specified in Materials and methods.
Regarding both the chilling (+4 °C for 0-16 days) and freezing (-4 °C for 0-30 hours) treatments, the kind of growth chambre used and other treatment conditions such as light irradiance and photoperiod applied for the chilling and freezing treatments have to be added.
In Results, LT50 values are given for GJ and GJ2501 genotypes; however, no description of LT50 values determination is provided in Materials and methods. I thus think that a brief description of LT50 determination has to be added in Materials and methods.
For all software kinds mentioned in Materials and methods, e.g., Kobas or Goatools software, software version and web address or any other relevant reference have to be given.
Conclusion: I think that the main message of the present study, i.e., how CuELIP1a overexpression contributes to enhanced cold tolerance of transgenic Arabidopsis should be added AS Figure 12 to the manuscript.
Formal comments on the text:
Introduction, line 43: Remove the words „and the citrus industry“ in the statement: „…which have caused great losses in the citrus industry, particularly for the late-maturing citrus that needs to survive the winter in the northern fringe growing areas.“
Results, line 160: Add the word „enhanced“ prior to „cold tolerance“ in the statement „These results implied that the enhanced cold tolerance of GJ2501 might be attributed to a higher number of cold-responsive DEGs relative to GJ.“
Results, line 173: Correct the typing error in the term „chlorophyll“ (not „chlorophyII“).
Results, line 228: Add a comma both before and after the word „respectively“ in the statement „These results implied that the DEGs in these unique GO terms, respectively, in GJ and GJ2501might also contribute to their differences in cold tolerance.“
Line 298: Replace the word „higher“ with ůenhanced“ in the statement „The enhanced expression of 12 up-regulated cold-responsive DEGs…“
Line 317: Add a comma both preceding and following the word „respectively“ in the statment „I tis noteworthy that CuKCS19 and CuKCS6 were, respectively, down-regulated by 100 and 845 folds in GJ…“
Line 336: Modify the statement as follows: „Taken together, these results indicated that overexpression of CuELIP1a from GJ2501 improved cold tolerance of Arabidopsis leading to enhanced POD activity and decraesed MDA content.“
Line 366: Replace „less“ with „lower“ in the statement „lower accumulation of MDa“.
Line 395: Replace the word „higher“ with „enhanced“ in the statement „…that enhzanced photoprotective capacity conferred by cuELIP1 in GJ2501 may contribute to its cold tolerance.“
Conclusion, line 474: Replace the word „less“ with „lower“ in the statement „Our work revealed that GJ2501 is more cold-tolerant than GJ, as evidenced by its lower photoinhibition and PSII photodamage, lower accumulation of MDA, but higher POD activity…“
Final recommendation: Reconsider after a major revision.
Reviewer 2 Report
Comments and Suggestions for Authors
I am very glad the authors wrote this manuscript. It is a well-written, needed, and useful . The authors are analytical and the tables/shcematics that they are presenting indeed help the author very much. I belive that the manuscript could be published prior to some minor revision. And after the authors have checked again some minor typos that exist. Also the fonds of some figures should be increased in size.
Research questions are well defined and within the aims and the scope of the journal. The introduction is adequate and includes in suitable way the relevant earlier publications. Materials are almost properly described. Methods are also almost properly described and used in a way that is possible to replicate. The investigation is performed to good technical standards. It is no ethical problem involved. A nicely conducted research (although a bit complicated) with conclusions well supported by the results. The level of English is adequate. Moreover, the selection of the plant species test needs to justified. Also please use uniform letter fonts.
Reviewer 3 Report
Comments and Suggestions for Authors
Why put the gene in the title? It might be better to include light-induced early protein or the role of the photosystem.
Well readable article. Clear study objectives.
Well written introduction, extensive and full of bibliographical references but with few details regarding the experimental approach used in the study.
Line 98, specify MDA in full at least once
Clear results, well explained and valid for understanding the article. Numerous and rich figures and tables.
Line 236, What each module includes, unclear
Good discussion, in line with the literature. Concise materials and methods.
Line 432-437 How was total RNA extracted and how was it quantified? How was the mRNA isolated? How was the cDNA library constructed? How were the reads cleaned? How was the PCA performed?
Interesting and useful study given the global importance of mandarin cultivation.
Line 577 2024? Mistake?
Comments on the Quality of English LanguageMinor editing of English language required
